# Emission Characteristics of Polycyclic Aromatic Hydrocarbons and Nitro-Polycyclic Aromatic Hydrocarbons from Open Burning of Rice Straw in the North of Vietnam

**DOI:** 10.3390/ijerph16132343

**Published:** 2019-07-02

**Authors:** Chau-Thuy Pham, Yaowatat Boongla, Trung-Dung Nghiem, Huu-Tuyen Le, Ning Tang, Akira Toriba, Kazuichi Hayakawa

**Affiliations:** 1Faculty of Environment, Vietnam National University of Agriculture, Hanoi 131001, Vietnam; 2Faculty of Science and Technology, Thammasat University, Pathumtani 12121, Thailand; 3School of Environmental Science and Technology, Hanoi University of Science and Technology, Hanoi 112400, Vietnam; 4VNU-University of Science, Vietnam National University-Hanoi, 334 Nguyen Trai, Hanoi 120000, Vietnam; 5Institute of Nature and Environmental Technology, Kanazawa University, Kakuma-machi, Kanazawa, Ishikawa 920-1192, Japan; 6Institute of Medical, Pharmaceutical and Health Sciences, Kanazawa University, Kakuma-machi, Kanazawa, Ishikawa 920-1192, Japan

**Keywords:** rice straw (RS), open burning, polycyclic aromatic hydrocarbon (PAHs), nitro-polycyclic aromatic hydrocarbon (NPAHs), the north of Vietnam

## Abstract

This research investigated the distribution and contribution of polycyclic aromatic hydrocarbons (PAHs) and nitro-polycyclic aromatic hydrocarbons (NPAHs) bound to particulate matter (PM) emitted from open burning of rice straw (RS) into the atmosphere in the north of Vietnam. The experiments were conducted to collect PM_2.5_ and total suspended particulates (TSP) prior to and during burning in the period of 2016–2018 in suburban areas of Hanoi. Nine PAHs and 18 NPAHs were determined using the HPLC-FL system. The results showed that the proportion of RS burning seasonally affects the variation of PAHs emission in atmospheric environment. The levels of nine PAHs from RS burning were 254.4 ± 87.8 µg g^−1^ for PM_2.5_ and 209.7 ± 89.5 µg g^−1^ for TSP. We observed the fact that, although fluoranthene (Flu) was the most abundant PAH among detected PAHs both in PM_2.5_ and TSP, the enrichment of Flu in TSP from burning smoke was higher than that in PM_2.5_ while the contribution of benzo[*a*]pyrene (B*a*P) and indeno[*1,2,3- cd*]pyrene (IDP) in PM_2.5_ from burning smoke were much higher than those in TSP. This research found that 1-nitropyrene (1-NP) and 6-nitrochrysene (6-NC) emit from RS burning with the same range with those from wood burning. The 2-nitrofluorene (2-NF) and 2-nitropyrene (2-NP) released from RS burning as the secondary NPAHs. This research provides a comprehensive contribution characterization of PAHs and NPAHs in PM with different size emitted from traditional local rice straw burning in the north of Vietnam. The results help to clarify the environmental behavior of toxic organic compounds from RS burning in Southeast Asia.

## 1. Introduction

Polycyclic aromatic hydrocarbons (PAHs) are highly lipid-soluble and mainly originate from imperfect combustion of organic matter and fossil fuel, for example, coal and biomass burning, internal combustion engines, heat and power generation and refuse burning. Nitro-polycyclic aromatic hydrocarbons (NPAHs) are a group of organic compounds formed of two or more condensed benzene rings linked by one or two nitro groups. NPAHs not only come from combustion of organic matter but also from the reaction of their parent PAHs with OH or NOx radicals in the atmosphere [1] and/or the heterogeneous gas-particle interaction of the parent PAHs adsorbed onto particulates with nitrating agents [2]. PAHs are considered to be the main causes of indirect-acting mutagenicity and higher carcinogenicity, while NPAHs are considered to be the main causes of the direct-acting mutagenicity [3,4]. Several NPAHs, such as, 1,3-, 1,6-, and 1,8-dinitropyrenes (DNPs) and 3-nitrobenzanthrone (NBA), show very strong direct-acting mutagenicities as compared to PAHs [5,6]. The mutagenic properties and potential carcinogenicity for some PAHs and NPAHs have been documented in the report of the International Agency for Research on Cancer (IARC) [7]. For examples, according to IARC, Benzo[*a*]pyrene (B*a*P) belongs to Group 1 (carcinogenic to humans), 1-nitropyrene (1-NP) belongs to Group 2A (probably carcinogenic to human) and several other PAHs and NPAHs, such as benzo[b]fluoranthene, benzo[k]fluoranthene and 6-nitrochrysene, are in Group 2B (possibly carcinogenic to humans).

Vietnam is one of the largest rice exporters in the world, 5th in global rice production, producing over 42.7 million tons paddy rice in 2017 according to Vietnam Statistics Portal, 2017 [8]. As a result, large amounts of crop residues primarily rice straw and rice stubble are produced. The total amount of rice straw (RS) generated in Vietnam is approximately 24 million dry tons if the proportion of RS subject to open field burning was 75% (calculated by Gadde et al., 2009 [9]). In which the Red River Delta and Mekong River Delta have higher density of RS than those in other regions due to high ratio between rice yield and rice-planted area [8]. Investigations in some case studies showed that 87% and 76% of the total RS were burned in the field after harvest in Mekong and Red River Deltas, respectively [10,11,12]. This practice introduces a considerable amount of pollutants into the atmosphere, the major compounds being CO_2_, CO, particulate matter (PM) and other organic compounds. Among these pollutants, PAHs or NPAHs exist in gas phase or particulate phase depending on their vapor pressure in the ambient air as constituents of the PM. Although they are released at low concentration, they have harmful toxicological properties, which have been linked to skin, lung, bladder, liver and stomach cancers. Health studies have reported an increase in bronchial asthma in children living close to rice fields during burning seasons [13,14].

The emission of PAHs from carbonaceous fuel combustion depends largely on the combustion conditions and the properties of different fuels [15,16]. Several researches reported the influence of combustion parameters (moisture content of fuel, temperature, oxygen supply) on PAHs emission level [17,18,19,20]. RS burning release more particles and also PAHs compared with other biofuel burning such as wheat straw, almond straw [19,21]. Jenkins et al. [22] found that the open burning of barley straw and wheat straw from wind tunnel simulations emitted much higher levels of PAHs, including B*a*P, than other cereal and wood fuel types burning. Less volatile five- and six-ring PAH was predominately on smaller particles and appeared in the early stages of combustion, while the more volatile three- and four-ring PAH formed on larger particles as the smoke cooled [19].

Recently, a number of RS burning experiments for the purpose of smoke emission measurement have been conducted both from open burning experiment and laboratory experiment in Southeast Asian countries such as Thailand [23,24] and China [18,25,26,27]. Some reports conducted on the effect of RS burning on atmospheric PAHs in Taiwan and China [28,29,30]. Open chamber burn simulation of RS burning also conducted in Spain to report emission profiles during controlled RS burning [20]. These researches measured various chemical compounds, whereas some investigations focused on selected chemical species such as PAHs, levoglucosan. However, measurement of Nitro-PAHs in particulate matter emitted from RS burning has not yet reported, although these compounds are more toxic than PAHs as mentioned above. 

Despite the fact that there are many governmental activities to minimize the fact of RS burning, open RS burning after harvest is still a common method in Vietnam due to the large amount of RS produced and it is easy to clean up the field afterwards. Up to now, the published information about emission from RS burning in Vietnam has been limited [31]. There has been no report on the organic composition of PM emitted from RS burning in Vietnam. This study was conducted to provide a comprehensive compositional characterization and the profile of PAHs and NPAHs emission from real-world of RS burning in the north of Vietnam. The contribution of PAHs and NPAHs from RS burning smoke to the atmospheric particles was examined in order to better understand the emission characteristics of RS open burning. The comparison from this research should help to clarify environmental behaviors of toxic organic compounds from RS burning in Southeast Asian countries.

## 2. Materials and Methods 

### 2.1. Experimental Design

All experiments were conducted at harvested paddy fields in the North of Vietnam during the RS burning seasons (October to June next year) in the period of 2016–2018 with 14 sampling sites in suburban areas around the Hanoi capital. The tests for open field burning experiments were designed in the same way as the common RS burning practices in order to characterize the emission of local RS burning in the north of Vietnam. Recently, the use of combine harvesters has been widely adopted by farmers to harvest mature rice. The upper part of RS was cut out in the field by combine harvesters while the lower part virtually untouched. Farmers usually collect RS into small piles after natural drying on the field and then burn them to clean the field and prepare area for the next crop. 

Field experiments were selected in the middle of agricultural field and far away from road and residential area to minimize the effect of local sources. Both background sampling (before burning, BG samples) and RS burning smoke sampling (during burning, BB samples) were conducted to identify the contribution of emission from RS burning to background environment. The BG sampling was conducted prior to burning and taken for a period of 2 h. BG and BB samples of each sampling site were conducted at the same day. After BG sampling was finished, the smoke plume sampling was taken immediately for a period of 20–30 min during burning. The details of sampling information were described in Appendix A (for PM_2.5_) and Appendix A (for TSP).

All sampling devices and continuous reading instruments were placed at a fixed downwind site in each experimental field, about 5 m away from downwind edge of the burning paddy, to avoid damage from the flame and heat and to be enough to catch the smoke from burning, follow the method from Kim Oanh et al., 2011 [23]. The PM samplers were located at about 1–1.5 m from each other, close enough to catch the same part of a smoke plume and far away enough to minimize the inlet flow disturbance. The sampling inlets were positioned at 1.5 m above ground level (Figure 1).

The total suspended particulates (TSP) were collected on quartz fiber filters (2500QAT-UP, 8 × 10 in, Pall Life Sciences, Ann Arbor, MI, USA) using a 120H Staplex high-volume air sampler at a flow rate of 1000 L min^−1^. PM_2.5_ samples were collected on the filters (2500QAT-UP, 47 mm ϕ, Pall Life Sciences, Ann Arbor, MI, USA) by mini volume sampler TAS, Airmetrics, USA (TAS-5.0, 4998) at a flow rate of 5 L min^−1^. Wind, temperature and humidity were recorded by continuous field measurement instrument (Madel Kestral 4000, Nielsen Kellerman, USA). The filters were kept in desiccators at room temperature within 48 h and weighed before and after sampling. Each filter was wrapped in aluminum foil and put in a sealable plastic bag and stored in a refrigerator at −20 °C until analysis. 

### 2.2. PAHs and NPAHs Analysis

#### 2.2.1. Reagents and Chemicals

The US.EPA 610 PAHs mix, a mixture of 9 PAHs (10 μg/mL in acetonitrile) including fluoranthene (Flu), pyrene (Pyr), benz[a]anthracene (B*a*A), chrysene (Chr), benzo[*b*]fluoranthene (B*b*F), benzo[*k*]fluoranthene (B*k*F), benzo[*a*]pyrene (B*a*P), dibenz[*a,h*]anthracene (DBA) and indeno[1,2,3- *cd*]pyrene (IDP), were purchased from Sigma-Aldrich (St. Louis, MO, USA). Three PAHs, pyrene-*d*_10_ (Pyr-*d*_10_), benzo[a]anthracene-*d*_12_ (B*a*A-*d*_12_) and benzo[*a*]pyrene- *d*_12_ (B*a*P-*d*_12_), used as internal standards for PAHs, were purchased from Wako Pure Chemicals (Osaka, Japan). 1,6- Dinitropyrene (1,6-DNP), 1,3-dinitropyrene (1,3-DNP), 1,8-dinitropyrene (1,8-DNP), 2-nitroanthracene (2-NA), 9-nitroanthracene (9-NA), 9-nitrophenathrene (9-NPh), 2-nitrofluorene (2-NF), 2-nitrofluoranthrene (2-NFR), 3-nitrofluoranthrene (3-NFR), 1-nitropyrene (1-NP), 7-nitrobenz[*a*]anthracene (7-NB*a*A), 6-nitrochrysene (6-NC) and 6-nitrobenz[*a*]pyrene (6-NB*a*P) were purchased from AccuStandard, Inc. (New Haven, CT, USA) (100 μg/mL in toluene). 1-Nitrofluoranthrene (1-NFR), 2-nitropyrene (2-NP), 1-nitroperylene (1-NPer) and 3-nitroperylene (3-NPer) were supplied from Chiron AS (Trondheim, Norway) (0.1 mg/mL in toluene), and 4-nitropyrene (4-NP) was supplied by Tokyo Chemical Industry (Tokyo, Japan). 6-Nitrochrysene-*d*_11_ (6NC-*d*_11_) was an internal standard for NPAH analysis and was purchased from Cambridge Isotope Lab. Inc. (Andover, MA, USA). All solvents and other chemicals were HPLC or analytical grade purchased from Wako Pure Chemical Industries, Ltd and Kanto Chemical Company (Tokyo, Japan).

#### 2.2.2. Extraction Procedure and PAHs and NPAHs Analysis

An area of 2.834 cm^2^ of the PM_2.5_ and 5.668 cm^2^ of TSP filters was cut into small pieces and put in 50 mL centrifugal tube. After the addition of internal standards, a mixture of B*a*A-*d*_12_ and B*a*P- *d*_12_ (60 and 33 ng/mL, respectively) for PAH quantification and 6-NC-*d*_11_ (10 ng/mL) for NPAH quantification, both PAHs and NPAHs on filter papers were ultrasonically extracted twice with 10 mL dichloromethane (DCM) for 15 min. After adding 60 μL of dimethyl sulfoxide (DMSO) to the extract, the DCM in the extract solution was completely evaporated. The PAHs and NPAHs in the residue was dissolved in 150 μL ethanol, the extract was filtered through a centrifugal membrane filter cartridge (Centricut, 0.2 μm). This step was repeated two times. Finally, an aliquot (110 μL) of the solution was injected into an HPLC with fluorescence detection (HPLC-FL). Ten PAHs and 18 NPAHs were determined using the HPLC-FL system as described according to our previous report [32].

## 3. Results and Discussions

### 3.1. The Distribution and Contribution of PAHs from RS Burning Smoke to the Atmospheric Particulates

#### 3.1.1. Distribution of PAHs in Background Environment

The sampling campaign was conducted in two rice harvest seasons (summer-autumn crop- harvested in October, 2016 and winter-spring crop- harvested in June, 2017, 2018) at 14 sampling sites in suburban areas of Hanoi. The results of PAHs in ambient air prior to burning are shown in Table 1 and Table 2. The concentration of total 9 PAHs in PM_2.5_ prior to RS burning varied from 0.89 to 18.5 ng m^−3^ with average value of 14.4 and 2.6 ng m^−3^ in autumn-winter and spring-summer season, respectively. The concentrations of PAHs in BG samples were distinguished in two harvest seasons. The mean concentration of nine PAHs adsorbed on PM_2.5_ in the autumn-winter season was seven times greater than that in spring-summer season. For TSP in BG samples, the concentrations of total nine PAHs ranged from 0.97 to 42.5 ng m^−3^, with an average value of 19.92 ng m^−3^ in the autumn-winter season, which was six times higher than the average value of total PAHs in the spring-summer season (3.41 ng m^−3^). 

The contribution of PAHs content in PM is useful information on the composition of PM. Hence, the PAHs amount absorbed on particles was calculated in µg g^−1^ PM for other view. The total amount of PAHs adsorbed on atmospheric particles in BG samples was fluctuated in the range of 4.7–34.8 µg g^−1^ for PM_2.5_ with average value of 25.3 µg g^−1^ in the autumn-winter season and 12.8 µg g^−1^ in the spring-summer season. There was small seasonal variation of the total amount of PAHs in PM_2.5_ and TSP collected from background samples, both in mass per total air volume (ng m^−3^) and mass per total aerosol mass (µg g^−1^). Although many previous studies found that the higher level of atmospheric PAHs in winter season than in summer season due to the changes of emission sources, the meteorological conditions or the secondary chemical reactions [33,34,35]. In the present study, the air temperature during sampling campaign varied from 25.8 to 34 °C in October, slightly different from those in June (varied from 31.7 to 36 °C). The small difference of temperature between two harvest seasons may not affect to the gas–particle partitioning and photo-degradation on governing the concentration of particulate PAHs. This sampling collected ambient particulate matter in suburban area, in which RS burning is the main source affected to the level of PAHs. We surveyed on 120 local farm householders of three suburban areas of Hanoi, where rice was the major crop and RS burning was common practice. The winter-spring crop (harvested in June) has the most favorable weather conditions for rice production in the year. However, after rice harvesting, it often rains in this season, leading to occasional burning of RS in June. With rainy days, households often bury straw in the field after harvest. The sunny weather in summer-autumn crop (harvested in October) is convenient for RS burning directly on the field. This also occurs in the Mekong river delta [36]. Results from the survey showed that the proportion of RS subject to open field burning in summer-autumn and winter-spring crop in suburban areas of Hanoi was about 70% and 40% of the total RS production, respectively. Then, the higher proportion of RS burning in summer-autumn harvest season may affect to PAHs level in the background atmospheric environment. Furthermore, rice harvest season in the north of Vietnam usually lasts 2–3 weeks depending on the maturity of rice. Hence, RS burning happens scattered during harvest season may affect to the quality of background atmospheric environment around the area during the RS burning season.

#### 3.1.2. Distribution of PAHs in RS Burning Smoke 

We determined nine targeted PAHs in both atmospheric particulates and RS burning smoke to find the contribution of RS burning to PAHs level in the atmosphere. The mean concentrations and standard deviation (SD) of individual PAHs for background environment and burning smoke in two sampling campaigns are shown in Table 1 and Table 2. The harvest seasonal difference was not observed for RS burning smoke. The mean concentrations of the nine PAHs in the summer-autumn and winter-spring crop were 4487 and 3064 ng m^−3^ for PM_2.5_ and 2700 and 2340 ng m^−3^ for TSP, respectively. The content of total PAHs adsorbed on PM (in mass per total aerosol mass) were 288.3 and 225.3 µg g^−1^ for PM_2.5_ and 246 and 183.3 µg g^−1^ for TSP in the summer-autumn and winter-spring crop, respectively. These values had not significant seasonal difference for both PM_2.5_ and TSP (P > 0.05). The substantial variation in the level of PAHs in volume of burning smoke (ng m^−3^) may be due to the different dilution ratios among sampling sites. The different dilution ratio of the smoke plume during field sampling related to the uncontrolled wind conditions, which is inevitable because one of the purposes of this research was to describe the real emission from local traditional practice of open RS burning in the north of Vietnam. 

The average content of total PAHs adsorbed on particulates from burning smoke in two sampling campaigns were 254.4 ± 87.8 µg g^−1^ for PM_2.5_ and 209.7 ± 89.5 µg g^−1^ for TSP. The results of PAHs in PM_2.5_ from this research were compared with those from Kim Oanh et al., 2011 [23], with the same open field burning experiment, but with different rice varieties and local burning practice. The comparison showed that the ratio of individual PAHs in PM_2.5_ in the present study was about 10 times smaller than those in the previous report [23]. This fact may relate to RS composition, moisture and/or burning type (spread or pile burning). The lower carbon content of RS in the present study (39.4 ± 7%) as compared to the value of RS in Thailand (49%) [37] may lead to smaller PAHs emission. Different fuel composition lead to different organic compounds adsorbed in PM [15,16]. Small pile RS burning is common practice in the north of Vietnam because of small fields area is convenient for RS gathering. RS often were gathered easily in several small piles on each field in order to burn more quickly after drying. The common burning type in Thailand was spread burning in a large scale and take places 2–7 days following the harvest season [23]. Spreading burning and pile burning associated to different RS moisture or burning temperature may lead to different PAHs emission. However, further study will need to be conducted to compare the emission profile from different rice composition in the control experiment (hood experiments) and to help clarify the variation of PAHs emission among rice varieties.

#### 3.1.3. Contribution of PAHs from RS Burning Smoke to Atmospheric Particulates

Mean concentrations of individual PAH from background samples and burning smoke in PM_2.5_ and TSP (ng m^−3^) are shown in Figure 2 and Figure 3. In general, the average concentrations of PAHs in burning smoke were significantly higher than those in BG samples; more than 100 times higher for most compounds in both PM_2.5_ and TSP. The levels of four-ring PAHs (Flu to Chr) in smoke had enrichment of above 400–1200 times as compared to those in PM_2.5_ from BG samples. In particular, the average concentration of B*a*A was 1200 times and of Chr was 715 times higher in BB samples in comparison with BG samples for PM_2.5_. The concentrations of high molecular weight PAHs with five rings and six rings in burning smoke did not increase as much as four rings PAHs, except B*a*P. Among the nine PAHs, Flu showed the highest concentration both in PM_2.5_ and TSP. However, the accumulation of Flu in TSP from burning smoke much higher than that in PM_2.5_ (978 times for TSP and 484 times for PM_2.5_ greater in BB samples as compared to BG samples). On the contrary, the accumulation of B*a*P and B*a*A in PM_2.5_ was higher than that in TSP. The average concentrations of B*a*P and B*a*A in burning smoke were 726 and 1200 times higher than that in BG samples for PM_2.5_ but only 116 and 192 times for TSP, respectively. 

Take a look at the difference of PAH ratio adsorbed on particles (µg g^−1^ PM) between background and burning samples. The results were calculated in average concentrations from two sampling campaigns. The results showed that the average concentration of B*a*A and B*a*P in PM_2.5_ from burning smoke were 28 and 27 times higher than those in BG samples, respectively. Followed by Chr, its average concentration was 21 times higher than those from BG samples. For TSP, the average concentrations of Fu, Pyr and B*a*A in burning smoke were 16, 7 and 6 times higher than those from background samples, respectively. The contribution of individual PAH in total accumulated PAHs in PM was showed in Figure 4. Here, each accumulated PAH is calculated as the difference between the content of PAH adsorbed on PM in BB samples (mg g^−1^) and the content of corresponding PAH in BG samples. The Flu contributed the highest percentage among nine detected PAHs in both PM_2.5_ and TSP emitted from RS burning (22.1 and 33.1%, respectively), followed by Pyr. The predominance of Flu and Pyr in particles emitted from RS burning in Hanoi was in accordance with the previous researches [24,27,28]. Despite the fact that Flu was the most abundant among detected PAH both in PM_2.5_ and TSP, the enrichment of Flu in TSP from burning smoke was higher than that in PM_2.5_, while the accumulation of B*a*P and IDP in PM_2.5_ from burning smoke were much higher than those in TSP (Figure 4). The mean contents, median values and content variations of individual PAH in PM_2.5_ from background environment and burning smoke were also showed in Figure 5a,b. These results indicated that the higher molecular PAHs such as B*a*P and IDP preferred concentrating on fine PM than TSP emitted from RS burning smoke. This result provided a significant variation of preferred contribution of PAHs emitted from RS burning, including B*a*P in PM_2.5_ and Flu in TSP. These results were in accordance with the previous reports [19,38,39], higher molecular PAHs more preferring concentrate on fine particulates than course particulates. This also occurred in the present study with particulates emitted at sources (RS burning). 

Furthermore, we examined the concentration ratios of PAHs relative to B*k*F in BG and BB samples to observe the contribution of PAHs from RS burning smoke to atmospheric pollution. We selected a relatively stable PAH, B*k*F, to compare the relative ratios of each PAHs to B*k*F [40]. Table 3 summarized PAHs profiles relative to B*k*F in PM_2.5_ and TSP in both BG and BB samples. The results showed that the concentration ratio of B*a*P/B*k*F in PM_2.5_ from burning smoke (9.14 ± 6.6) was much higher than that in BG samples (1.2 ± 0.7). This ratio is smaller in TSP from burning smoke (3.8 ± 1.9) and only about three times higher than that in TSP from BG samples (1.08 ± 0.65). Among these ratios in PM_2.5_, the concentration ratio of Flu/B*k*F in RS burning smoke was the highest value. However, the concentration ratios of B*a*A, Chr and B*a*P relative to B*k*F in PM_2.5_ from RS burning smoke was significantly higher (P < 0.001) as compared to those in BG samples, especially the average concentration ratio of B*a*P/B*k*F in burning smoke, which was 7.43 times greater than that in background samples (P < 0.001), while the concentration ratios of Flu, Pyr, DBA and IDP relative to B*k*F was not significantly higher in comparison with the BG samples (P > 0.02). These results are further evidence to indicate the considerable contribution of B*a*P, B*a*A, Chr in PM_2.5_ and Flu in TSP from RS burning smoke. B*a*P, B*a*A and Chr was more toxic, stable and less volatile than Flu. This fact is in accordance with previous findings from agricultural burning [19]; less volatile PAH was predominately on smaller particles while the more volatile PAH formed on larger particles.

The concentration ratios of PAHs relative to B*k*F in PM_2.5_ in the present study were compared with those from Kim Oanh et al., 2011 [23]. The results showed that among all calculated ratios, the concentration ratios of B*a*P/B*k*F and B*a*A/B*k*F were much higher than those in the previous research (3.5 and 4.8 times greater, respectively). Although there was a lower contribution of PAHs in PM, the concentration ratio of PAHs relative to B*k*F in this study was higher than that from previous research. This fact may relate to residues in RS from pesticide and/or fertilizer use during rice cultivation and/or different rice varieties. To clear this variation, other experiments will need to be conducted.

### 3.2. The Distribution and Contribution of NPAHs in Particulates Emitted from RS Burning

#### 3.2.1. Observation of Primary and Secondary NPAHs from RS Burning

NPAHs could be formed from the reaction of the parent PAHs with gaseous nitrogen oxides during not only incomplete combustion of organic matter but also transportation in the atmosphere [1,6,41]. This is the first study reporting NPAH concentrations from RS burning smoke to fill the gap of NPAHs derived from RS burning. We reported NPAHs concentrations in PM_2.5_ and TSP from background atmospheric environment and RS burning smoke in Table 4. Several NPAHs were observed in our sampling. However, their concentrations were very low. The concentration of 1-NP in PM_2.5_ ranged from 0.01 to 0.79 µg g^−1^ with the average value and SD of 0.34 ± 0.31 µg g^−1^ in BG samples. This compound was smaller in PM_2.5_ from burning smokes (0.05 ± 0.03 µg g^−1^). Among detected NPAHs, the seasonal variation was observed clearly with 2-NF both in BG and BB samples for PM_2.5_ and TSP. The concentration of 2-NF in PM_2.5_ in autumn-winter harvest season (6.2 ± 2.1 and 7.1 ± 4.9 µg g^−1^ in BG and BB samples, respectively) was higher than that in spring-summer one (0.5 ± 0.7 and 0.09 ± 0.01 µg g^−1^ in BG and BB samples, respectively). This trend is similar for TSP. This result suggested the stronger formation of 2-NF in autumn-winter crop should be considered. The concentration of 2-NF in PM_2.5_ ranged from 0.97 to 8.8 µg g^−1^ with the average value and SD of 5.3 ± 2.8 µg g^−1^ in BG samples, which was not significantly different from those in BB samples (0.08 to 12.3 µg g^−1^, P > 0.05) with an average value and SD of 5.07 ± 5.22 µg g^−1^. The concentration of 2-NP in PM_2.5_ from burning smoke (0.76 ± 0.74 µg g^−1^) was slightly smaller than in BG samples (2.7± 2.9 µg g^−1^). The concentration of 2-NP was similar between BG samples (0.46 ± 0.42 µg g^−1^) and BB samples (0.54 ± 0.46 µg g^−1^) for TSP. It has been known that, 2-NF and 2-NP are secondary NPAHs formed from gas phase reaction involving OH or NO_3_ radicals [1,41,42]. These results indicated that the secondary NPAHs (2-NF and 2-NP) could be formed in a secondary reaction between mother PAHs and radical agents in certain time and their concentrations were not significantly different between atmospheric particles and RS burning smoke. The higher level of 2-NF as compared to 2-NP was appropriate with the findings from the previous research [43]. This research found that yield of 2-NF from reaction of their parent PAHs with OH and NO_3_ radicals (in the present of NOx) was approximately 3% and 24%, respectively, and yield of 2-NP was approximate only 0.3% and 0.06%, respectively. 

In this sampling campaign, 6-NC was not detected in PM_2.5_ from BG samples but it was observed in RS burning smoke with the average value and standard deviation of 0.09 ± 0.06 µg g^−1^ for PM_2.5_ and 0.04 ± 0.03 µg g^−1^ for TSP. This fact suggested that 6-NC can emit from RS burning. Nitro-PAHs such as 6-NC was observed in the atmosphere, both from original source such as diesel engine [41], coal burning and wood burning [6] and from reaction of the parent PAHs in the atmosphere [44,45]. The average level of 6-NC in TSP from RS burning (0.04 ± 0.03 µg g^−1^) was almost the same as that of wood burning (0.05 ± 0.08 µg g^−1^) and slightly lower than that of coal burning (0.11 ± 0.19 µg g^−1^). Data of 6-NC from coal burning and wood burning were obtained from the previous report [6]. The results of 6-NC indicated that RS smoke may have adverse effect to human health because 6-NC belongs to group 2B (IARC), is possibly carcinogenic to humans. These results provide a useful information base for future RS burning management. The mean, median value and variation on the content of individual NPAH in PM_2.5_ from background atmospheric environment and RS burning smoke were described in Figure 6.

#### 3.2.2. The Concentration Ratio of NPAH/PAH in RS Burning

The concentration ratios of NPAHs/PAHs in this observation were described in Table 4. The concentration ratio of 2-NF/1-NP have been used to further identify the relative contribution of primary source or atmospheric gas phase formation source of NPAHs [2,45,46,47]. The ratio of [2-NF]/[1-NP], which are greater than 5, suggests the importance of gas-phase formation of NPAHs [45]. In this study, the [2-NF]/[1-NP] ratio was greater than 5 in almost sampling sites, which indicates the relative importance of secondary formation of NPAHs. Furthermore, this ratio in autumn-winter harvest season was significantly greater than spring-summer harvest season both in BG and BB samples for PM_2.5_ and TSP (Table 4). This result suggested that the formation of 2-NF was more favorable in autumn-winter crop than spring-summer crop. This fact may be due to the higher amount of OH and/or NO_3_ radicals in autumn-winter crop as compared to spring-summer crop.

Other concentration ratio of NPAH/PAH was used to identify the main sources of NPAHs. The average value and standard deviation of [1NP]/[Pyr] ratio in PM_2.5_ from atmospheric particles before burning was 0.095 ± 0.086 (Table 4). This value was reduced dramatically in PM_2.5_ emitted from burning smoke (0.0019 ± 0.0024) (Table 4). The values of [1NP]/[Pyr] ratio for RS burning were close to the values for coal burning (referenced from Tang et el., 2005 [48]). The temperature of RS burning bulk measured in this study varied from 300 to 700 °C depending on upper, middle or bottom part of the bulk. RS burning temperature was slightly smaller than coal burning temperature (about 900 °C). This fact suggests that the ratio of [NPAHs]/[PAHs] in particulates from RS burning might be in the same range as those from coal burning. This result was in agreement with the previous study, where lower fuel burning temperature, a lower yield of NPAHs from the corresponding PAHs and a lower [NPAH]/[PAH] ratios were observed [6,48]. 

The levels of 1-NP and 6-NC in TSP from RS burning smoke were compared with those from motorcycle exhausts. Data from motorcycle exhausts were obtained from our previous report [49]. The profile of NPAHs from RS smoke and motorcycle exhausts are shown in Figure 7. The levels of NPAHs from motorcycle exhaust were considerable higher than those from RS burning smoke. The mean concentrations of 1-NP and 6-NC from motorcycle exhausts were 17 and 114 times higher than those from RS burning smoke. The exhaust patterns of 1-NP and 6-NC were different between motorcycle exhausts and RS smoke. The level of 1-NP in RS burning smoke was considerable higher than that of 6-NC. However, the opposite trend was exhibited with motorcycle exhaust. In addition, the NPAHs concentrations in this research were compared with those from coal burning and wood burning. Data from coal burning and wood burning were calculated from the previous report [6]. The average level of 1-NP in TSP from RS burning in this study (0.18 ± 0.14 µg g^−1^) was in the same range with those from wood burning and coal burning (0.27 ± 0.44 µg g^−1^ and 0.27 ± 0.17 µg g^−1^, respectively). The comparison of 6-NC was observed as the same with 1-NP as described above. This suggested that the levels of NPAHs were in the same range as those from biomass burning and lower than those from non-biomass fuel burning. This result was in appropriate with the fact that 1-NP and 6-NC also is mainly come directly from diesel and gasoline engine exhaust [49,50] and the higher fuel burning temperature, the more NPAHs produces [6,48].

## 4. Conclusions

The total amount of nine PAHs bound to particulate matter in the background environment in suburban areas of Hanoi were different between two harvest seasons. The high proportion of RS subject to open field burning in summer-autumn harvest season contributed to the high level of PAHs in the atmospheric environment.The levels the 9 PAHs from RS burning were 254.4 ± 87.8 µg g^−1^ for PM_2.5_ and 209.7 ± 89.5 µg g^−1^ for TSP. Although Flu was the most abundant PAH among detected PAHs in both PM_2.5_ and TSP, the enrichment of Flu in TSP from burning smoke was higher than that in PM_2.5_, while the accumulation of B*a*P and IDP in PM_2.5_ from burning smoke were much higher than those in TSP. The considerable contribution of B*a*P and IDP in PM_2.5_ emitted from RS burning make more attention on the toxicity of PM_2.5_ from RS burning smoke, in which World Health Organization (WHO) ranked PM_2.5_ in Group 1 (carcinogenic to human). This result is useful information help to Vietnamese Government pay more attention to control the RS burning practice.The amount of PAHs in PM_2.5_ in the present study were 10 times lower than those from RS burning in Thailand, however the concentration ratio of PAHs relative to B*k*F were higher in comparison to the results from RS burning in Thailand. This variation may relate to rice composition and need to have further study.The 1-NP and 6-NC was observed from RS burning. However, their concentrations were in the same range with those from wood burning and much smaller than those from motorcycle exhausts. The 2-NF and 2-NP were observed as the secondary NPAHs in this sampling, in which the formation of 2-NF is more favorable than that of 2-NP. The concentration ratio of 2-NF/1-NP in this sampling was greater than five, which also suggested the relative importance of the gas-phase formation of 2-NF.The concentration ratio of mono NPAH to its mother PAH ([1-NP]/[Pyr]) from RS burning was close to the range from coal burning. This fact may associate with the same range of burning temperature between RS burning and coal burning. This research helps to clarify the emission characteristics and the environmental behavior of PAHs and NPAHs from open burning of RS in the north of Vietnam as well as in Southeast Asia.

## Figures and Tables

**Figure 1 ijerph-16-02343-f001:**
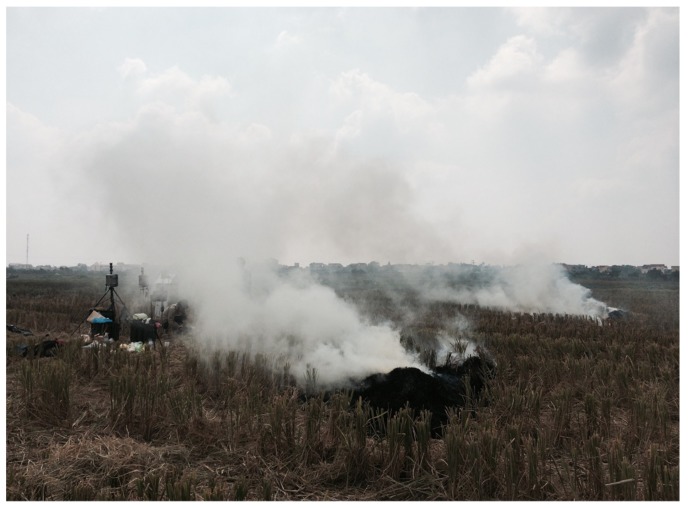
Field experiment for open burning of rice straw (RS).

**Figure 2 ijerph-16-02343-f002:**
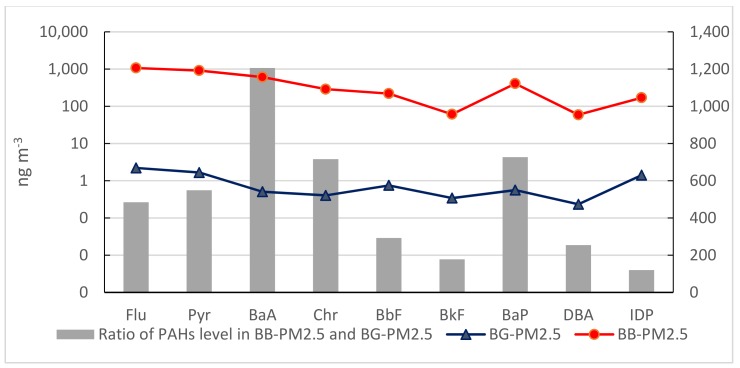
Comparison of PAH levels ^1^ in PM_2.5_ between BG ^2^ and BB ^3^ samples. ^1^ The values in Figure 2 and Figure 3 were the mean values of individual PAH concentrations (ng m^−3^) from BG and BB samples. ^2^: Background samples, ^3^: RS Burning samples

**Figure 3 ijerph-16-02343-f003:**
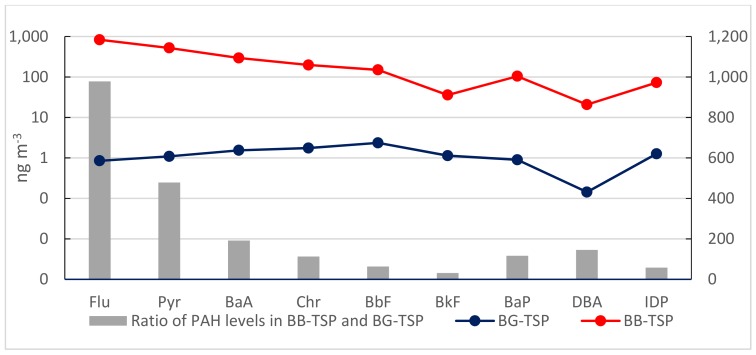
Comparison of PAH levels ^1^ in TSP between BG ^2^ and BB ^3^ samples. ^1^ The values in Figure 2 and Figure 3 were the mean values of individual PAH concentrations (ng m^−3^) from BG and BB samples. ^2^: Background samples, ^3^: RS Burning samples

**Figure 4 ijerph-16-02343-f004:**
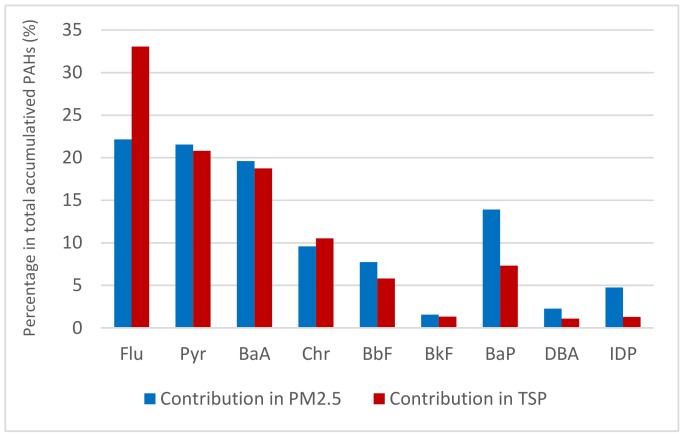
Contribution percentage^1^ of individual PAH in PM_2.5_ and TSP. The percentage of each PAH in total accumulated PAH in Figure 4 were calculated as the difference between the content of PAH adsorbed on PM in BB samples (mg g^−1^) and the content of corresponding PAH on PM in BG samples.

**Figure 5 ijerph-16-02343-f005:**
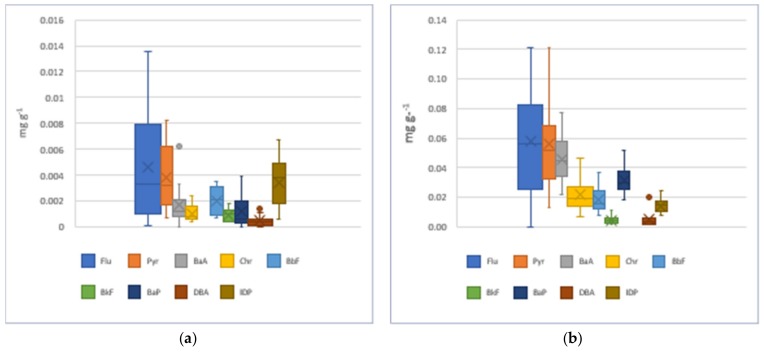
Distribution of individual PAH in PM_2.5_: (**a**) background samples; (**b**) RS burning samples. X marker in each bar chart: the mean of PAH content. Dash in each bar chart: the median of PAH content.

**Figure 6 ijerph-16-02343-f006:**
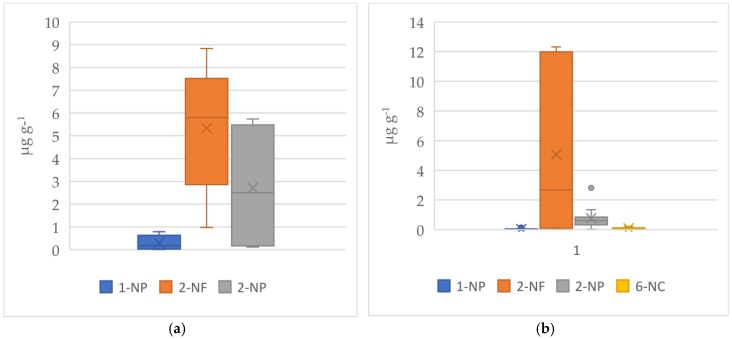
Distribution of individual NPAH in PM_2.5_: (**a**) background samples; (**b**) RS burning samples. X marker in each bar chart: the mean of NPAH content. Dash in each bar chart: the median of NPAH content.

**Figure 7 ijerph-16-02343-f007:**
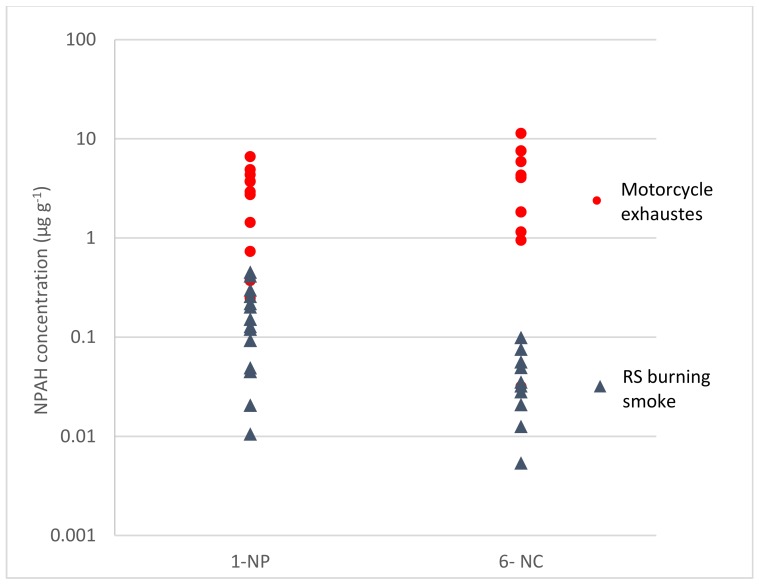
Comparison of NPAH levels ^1^ in TSP between rice straw burning smoke and motorcycle exhausts. ^1^ The level of individual NPAH in TSP was showed in µg g^−1^. RS burning smoke (n = 14); motorcycle exhausts (n = 10).

**Table 1 ijerph-16-02343-t001:** Concentrations of PAHs in ng m^−3.^

PAHs (ng m^−3^)	BG ^1^, PM_2.5_		BB ^2^, PM_2.5_		BG, TSP		BB, TSP	
	Autumn-Winter	Spring-Summer	Autumn-Winter	Spring-Summer	Autumn-Winter	Spring-Summer	Autumn-Winter	Spring-Summer
Flu	4.46 ± 1.71	0.28 ± 0.34	1219.6 ± 1208.1	923.1 ± 911.2	1.52 ± 1.28	0.27 ± 0.20	735.5 ± 678.4	996.4 ± 788.9
Pyr	3.09 ± 0.84	0.44 ± 0.32	1166.0 ± 1150.5	695.8 ± 666.0	1.87 ± 1.22	0.43 ± 0.24	571.7 ± 499.2	605.0 ± 507.8
B*a*A	0.71 ± 0.57	0.34 ± 0.31	793.4 ± 566.3	455.1 ± 250.3	3.08 ± 2.99	0.22 ± 0.22	668.1 ± 572.3	164.9 ± 125.0
Chr	0.69 ± 0.37	0.16 ± 0.09	409.1 ± 292.2	185.7 ± 134.6	3.37 ± 2.95	0.37 ± 0.21	301.9 ± 219.7	177.8 ± 112.2
B*b*F	1.14 ± 0.55	0.42 ± 0.23	172.5 ± 132.4	261.0 ± 139.5	4.36 ± 3.31	0.64 ± 0.51	160.1 ± 91.0	158.8 ± 96.0
B*k*F	0.55 ± 0.29	0.17 ± 0.08	93.9 ± 75.2	32.9 ± 21.3	2.09 ± 1.53	0.32 ± 0.21	70.0 ± 42.8	21.4 ± 14.9
B*a*P	1.04 ± 0.80	0.15 ± 0.09	390.2 ± 289.7	424.2 ± 233.7	1.52 ± 1.11	0.37 ± 0.35	109.9 ± 63.6	110.9 ± 65.5
DBA	0.46 ± 0.18	0.05 ± 0.02	42.0 ± 42.7	74.1 ± 78.6	0.25 ± 0.22	0.05 ± 0.04	13.1 ± 10.5	28.3 ± 10.2
IDP	2.35 ± 1.23	0.61 ± 0.33	201.0 ± 151.6	144.0 ± 104.9	1.87 ± 1.08	0.75 ± 0.40	69.8 ± 68.1	77.3 ± 55.1
**Total**	**14.41 ± 3.69**	**2.60 ± 1.31**	**4487.6 ± 3850.1**	**3064.1 ± 2370.3**	**19.92 ± 15.37**	**3.41 ± 1.93**	**2700.1 ± 1965.3**	**2340.8 ± 1732.8**

^1^ Background samples, ^2^ RS Burning samples. Abbreviations: Flu: Fluoranthene; Pyr: Pyrene; B*a*A: Benz[*a*]anthracene; Chr: Chrysene; B*b*F: benzo[*b*]fluoranthene; B*k*F: Benzo[*k*]fluoranthene; B*a*P: Benzo[*a*]pyrene; DBA: Dibenz[*a,h*]anthracene; IDP: Indeno[1,2,3- *cd*]pyrene; PM_2.5_: Particulate matter with diameter < 2.5µm; TSP: Total suspended particulate.

**Table 2 ijerph-16-02343-t002:** PAHs content in PM (µg/g PM).

PAHs (µg g^−1^)	BG ^1^, PM_2.5_		BB^2^, PM_2.5_		BG, TSP		BB, TSP	
	Autumn-Winter	Spring-Summer	Autumn-Winter	Spring-Summer	Autumn-Winter	Spring-Summer	Autumn-Winter	Spring-Summer
Flu	7.91 ± 3.71	1.57 ± 1.94	71.28 ± 41.16	46.36 ± 31.87	5.29 ± 3.90	2.59 ± 1.96	54.18 ± 32.98	72.37 ± 36.80
Pyr	5.35 ± 2.03	2.21± 1.95	68.15 ± 40.91	44.97 ± 19.82	6.68 ± 3.59	4.10 ± 2.06	39.01 ± 16.28	42.28 ± 22.83
B*a*A	1.33 ± 1.13	1.76 ± 1.88	55.15 ± 13.19	37.48 ± 11.77	10.60 ± 8.93	2.15 ± 2.28	65.51 ± 40.82	15.20 ± 12.28
Chr	1.26 ± 0.74	0.76 ± 0.47	30.07 ± 9.27	14.88 ± 4.89	11.57 ± 8.50	3.45 ± 1.78	38.32 ± 37.21	16.25 ± 10.44
B*b*F	2.04 ± 1.13	1.98 ± 0.91	12.31 ± 4.29	22.89 ± 8.31	15.26 ± 10.08	5.99 ± 4.56	19.21 ± 12.32	15.22 ± 11.74
B*k*F	0.99 ± 0.57	0.80 ± 0.38	6.52 ± 2.32	2.74 ± 1.02	7.35 ± 4.67	2.97 ± 1.66	9.19 ± 7.30	1.82 ± 1.08
B*a*P	1.72 ± 1.68	0.68 ± 0.52	27.67 ± 8.91	35.26 ± 8.97	5.32 ± 3.13	3.39 ± 3.03	12.94 ± 5.78	10.15 ± 7.47
DBA	0.67 ± 0.50	0.19 ± 0.15	2.64 ± 1.62	6.66 ± 6.29	0.88 ± 0.65	0.50 ± 0.41	1.37 ± 0.66	3.46 ± 3.31
IDP	4.06 ± 2.26	2.87 ± 1.47	14.53 ± 5.77	14.04 ± 4.58	6.73 ± 3.01	7.18 ± 3.75	7.01 ± 3.58	6.59 ± 6.75
**Total**	**25.34 ± 9.15**	**12.81 ± 7.84**	**288.30 ± 114.76**	**225.28 ± 48.05**	**69.68 ± 44.97**	**32.33 ± 16.81**	**246.74 ± 72.94**	**183.34 ± 95.79**

^1^ Background samples, ^2^ RS Burning samples. Abbreviations: Flu: Fluoranthene; Pyr: Pyrene; B*a*A: Benz[*a*]anthracene; Chr: Chrysene; B*b*F: benzo[*b*]fluoranthene; B*k*F: Benzo[*k*]fluoranthene; B*a*P: Benzo[*a*]pyrene; DBA: Dibenz[*a,h*]anthracene; IDP: Indeno[1,2,3- *cd*]pyrene; PM_2.5_: Particulate matter with diameter < 2.5µm; TSP: Total suspended particulate.

**Table 3 ijerph-16-02343-t003:** Summary of PAHs profiles relative to B*k*F in PM_2.5_ and TSP.

**The Ratios of Each PAHs Relative to B*k*F for PM_2.5_**							
Ratios ^1^	Flu/B*k*F	Pyr/B*k*F	B*a*A/B*k*F	Chr/B*k*F	B*a*P/B*k*F	DBA/B*k*F	IDP/B*k*F
BG samples (the present study)							
	5.55 ± 6.88	4.97 ± 4.98	2.43 ± 2.70	1.15 ± 0.42	1.23 ± 0.76	0.48 ± 0.30	3.82 ± 1.13
BB samples (the present study)							
	14.84 ±1 1.0	13.08 ± 9.54	11.69 ± 9.77	5.11 ± 3.36	9.14 ± 6.60	2.11 ± 4.06	3.83 ± 1.74
BB samples (Kim Oanh et al., 2011) ^2^							
	10	5.8	2.4	3.4	2.6		
**The Ratios of Each PAHs Relative to BkF for TSP**							
Ratios ^1^	Flu/B*k*F	Pyr/B*k*F	BaA/B*k*F	Chr/B*k*F	B*a*P/B*k*F	DBA/B*k*F	IDP/B*k*F
BG samples (the present study)							
	0.80 ± 0.43	1.5 ± 0.97	1.14 ± 1.14	1.44 ± 0.51	1.08 ± 0.62	0.18 ± 0.15	2.01 ± 1.11
BB samples (the present study)							
	28.39 ± 0.93	15.74 ± 11.82	7.75 ± 2.32	5.81 ± 3.27	3.85 ± 1.92	1.14 ± 1.12	3.71 ± 2.75

^1^ All ratios were calculated from the PAHs content in PM (mg g^−1^ PM). ^2^ The relative ratios in Kim Oanh et al., 2011 were calculated from average values of PAHs in PM_2.5_ (mg g^−1^ PM) in RS burning smoke. Abbreviations: Flu: Fluoranthene; Pyr: Pyrene; B*a*A: Benz[*a*]anthracene; Chr: Chrysene; B*k*F: Benzo[*k*]fluoranthene; B*a*P: Benzo[*a*]pyrene; DBA: Dibenz[*a,h*]anthracene; IDP: Indeno[1,2,3- *cd*]pyrene.

**Table 4 ijerph-16-02343-t004:** NPAH concentrations and the NPAHs/PAHs ratios in PM_2.5_ and TSP.

NPAHs (µg g^−1^)	BG ^1^-PM_2.5_			BB ^2^-PM_2.5_		
	Autumn-Winter	Spring-Summer	Average ^1^	Autumn-Winter	Spring-Summer	Average ^3^
1-NP	0.51 ± 0.25	0.04 ± 0.03	0.34 ± 0.31	0.07 ± 0.04	0.03 ± 0.01	0.05 ± 0.03
2-NF	6.2 ± 2.1	0.5 ± 0.7	5.33 ± 2.84	7.1 ± 4.9	0.09 ± 0.01	5.07 ± 5.22
2-NP	2.06 ± 3.19	2.34 ± 3.3	2.71±2.92	0.24 ± 0.28	1.23 ± 0.77	0.76 ± 0.74
6-NC	ND ^4^	ND ^4^	ND ^4^	0.04 ± 0.02	0.15 ± 0.04	0.09 ± 0.06
**Ratios**						
1-NP/Pyr	0.105 ± 0.101	0.036 ± 0.03	0.095 ± 0.086	0.0032 ± 0.0032	0.0009 ± 0.0004	0.0019 ± 0.0024
2-NF/1-NP	196.5 ± 404.2	15.11 ± 21.37	168.83 ± 367.83	131.04 ± 80.07	2.93 ± 0.78	131.04 ± 80.07
**NPAHs (µg g^−1^)**	BG-TSP			BB-TSP		
1-NP	0.035 ± 0.027	0.018 ± 0.008	0.03 ± 0.02	0.18 ± 0.14	0.18 ± 0.15	0.18 ± 0.14
2-NF	3.73 ± 3.27	1.13 ± 1.12	1.99 ± 2.17	15.8 ± 12.82	0.02 ± 0.02	6.33 ± 10.76
2-NP	0.13 ± 0.195	0.74 ± 0.35	0.46 ± 0.42	0.27 ± 0.37	0.77 ± 0.42	0.54 ± 0.46
6-NC	0.047 ± 0.042	0.06 ± 0.049	0.06 ±0.05	0.03 ± 0.02	0.04 ± 0.02	0.04 ± 0.03
**Ratios**						
1-NP/Pyr	0.004 ± 0.002	0.003 ± 0.001	0.004 ± 0.002	0.005 ± 0.003	0.004 ± 0.002	0.004 ± 0.003
2-NF/1-NP	203.6 ± 253.9	45.6 ± 41.1	98.3 ± 143.4	171.7 ± 49.1	0.61 ± 0.45	69.1 ± 96.9

^1^ Background samples; ^2^ RS Burning samples; ^3^ The average values were calculated from all samples in sampling campaign; ^4^ Not detected. Abbreviations: 1-NP: 1-nitropyrene; 2-NF: 2-nitrofluorene; 2-NP: 2-nitropyrene; 6-NC: 6-nitrochrysene; Pyr: Pyrene.

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
