# Peer review of "Emission Characteristics of Polycyclic Aromatic Hydrocarbons and Nitro-Polycyclic Aromatic Hydrocarbons from Open Burning of Rice Straw in the North of Vietnam"

_ijerph, 2019, doi:10.3390/ijerph16132343_

Round 1
Reviewer 1 Report
BaP is used in the introduction but is not defined until in Materials and Methods
Some sentences are broken and difficult to understand some minor parts of the work.
Spectroscopic analysis (NMR ---) of the smoke from the RS could add more meaning to the data if enough smoke can be collected for NMR. Many PAHs have distinctive splitting patterns in the aromatic region. Could be outside the scope of this work but extracts from the dry and most/wet RS could be analyzed and a comparative analysis of the amounts of PAHs be done.
Author Response
Answer to the reviewer comments
Thank you very much for your kind comments. I have read the valuable comments carefully and revised my manuscript as follows. Please find revised text with red colors in the track changes in the revised manuscript.
1. BaP is used in the introduction but is not defined until in Materials and Methods
2. Some sentences are broken and difficult to understand some minor parts of the work.
3. Spectroscopic analysis (NMR ---) of the smoke from the RS could add more meaning to the data if enough smoke can be collected for NMR. Many PAHs have distinctive splitting patterns in the aromatic region. Could be outside the scope of this work but extracts from the dry and most/wet RS could be analyzed and a comparative analysis of the amounts of PAHs be done
[Answers]:
1Ã According to your comment, we have checked and defined BaP (full name and abbreviation) in the introduction part (line 77 in the track changes version).
2Ã Thank you very much for your comment. We have revised and checked English grammar through the manuscript.
3Ã Thank you very much for your valuable suggestion. Spectroscopic analysis (NMR ---) of the smoke from the RS is interesting method to add more meaning of data. However, this is out of our scope. Research on PAHs emission from dry and wet RS (different moisture of RS) were conducted in the previous reports (Korenaga et al., 2001, Lu et al., 2009). Next research we will consider about that point.
Thank you very much

Reviewer 2 Report
Since there are limited sampling data, please show the following data: air volumes (m3) sampled for TSP and PM2.5, collected weights (mg) of TSP and PM2.5, concentrations (mg m-3) of TSP and PM2.5 in BG and BB.
There are a lot of minor errors in the text, please read carefully and correct them. For example, in Abstract please show full spellings of PM2.5, TSP, NP, NC, and NF.
Author Response
Response to Reviewer 2 Comments
Thank you very much for your kind comments. We have read the valuable comments carefully and revised our manuscript as follows. Please find revised text with red colorsin the track changes in the revised manuscript.
Point 1.Since there are limited sampling data, please show the following data: air volumes (m3) sampled for TSP and PM2.5, collected weights (mg) of TSP and PM2.5, concentrations (mg m-3) of TSP and PM2.5 in BG and BB.
[Answers 1]:Thank you very much for your comments. We would like to show the air volumes (m3) sampled for TSP and PM2.5, collected weights (mg) of TSP and PM2.5, concentrations (mg m-3) of TSP and PM2.5 in BG and BB in supplementary data (you can see in the Appendix).
Point 2.There are a lot of minor errors in the text, please read carefully and correct them. For example, in Abstract please show full spellings of PM2.5, TSP, NP, NC, and NF.
[Answers 1]:We have checked careful through the manuscript and have added full spellings of NPAHs compounds in the abstract

Reviewer 3 Report
This is a very valuable research, since authors have measured for the first time NPAHs from rice straw burning in addition to PAH. Results are very interesting and useful, nevertheless I have doubts about the paper structure and NPAHs results do not correspond to the explained methodology. Then, I think this issue should be solved before publication.
Materials and methods
I understand that background samples were taken in the same site just before biomass burning, is this true? but maybe they were taken one or more days. Please clarify, since wind direction can be different every day.
My major concern is that in the methodology section it seems that PAH and NPAH were quantified in BG and BB samples of TSP and PM2.5, during the both seasons; nevertheless the discussion of PAH is very different that the NPAHs one.
In the case of PAHs the authors presented a discussion between TSP and PM2.5 and the PAH profiles during BG and BB in the two seasons, as well as the comparison of PAH ratios between TSP and PM2.5. But that is not the case for NPAHs where the quantification in TSP is missing, there is no comparison between the NPAH in the two seasons as is the case with PAH.
The reader would expect for NPAHs a similar Table than Table 2 related with PAH. But the discussion related to NPAHs is focused to each day of sampling, and the profile is not presented.
It is not clear if NPAH/PAH ratios were estimated in both season or only in one. Table 4 only presents the results of one season and only for PM2.5.
Line 394-395. Authors mentioned that “the levels of NPAHs from RS burning in this study were in the same range with those from wood burning and slightly lower than those from coal burning “, but they did not indicate the references of those studies.
Regarding the comparison with motorcycle emissions, I do not think that concentrations comparison is the best option since sampling methods are quite different. I understand the point of the authors which want to make a comparison about the NPAHs 1NP and 6NC, but maybe would be more convenient the emission profile comparison. Additionally, the comparison should be between TSP since that particle size was used for motorcycles, but it seems that it was made with RS PM2.5 emissions.
Author Response
Response to Reviewer 3 Comments
Thank you very much for your valuable comments. We have read your comments carefully and revised our manuscript as follows. Please find revised text with red colorsin the track changes in the revised manuscript.
Point 1:I understand that background samples were taken in the same site just before biomass burning, is this true? but maybe they were taken one or more days. Please clarify, since wind direction can be different every day.
[Answer 1]:Our experiments were conducted for BG and BB samples in the same site at the same day. The BG sampling was conducted prior to burning and taken for a period of 2 hours. After BG sampling was finished, the smoke plume sampling was taken immediately. So the wind direction and other meteorological condition should be same between BG and BB samples. We explained more clear in the text (lines 140-142, lines number following the track changes version)
Point 2: My major concern is that in the methodology section it seems that PAH and NPAH were quantified in BG and BB samples of TSP and PM2.5, during the both seasons; nevertheless the discussion of PAH is very different that the NPAHs one.
In the case of PAHs the authors presented a discussion between TSP and PM2.5 and the PAH profiles during BG and BB in the two seasons, as well as the comparison of PAH ratios between TSP and PM2.5. But that is not the case for NPAHs where the quantification in TSP is missing, there is no comparison between the NPAH in the two seasons as is the case with PAH.
The reader would expect for NPAHs a similar Table than Table 2 related with PAH. But the discussion related to NPAHs is focused to each day of sampling, and the profile is not presented.
It is not clear if NPAH/PAH ratios were estimated in both season or only in one. Table 4 only presents the results of one season and only for PM2.5.
[Answer 2]:We would like to focus the results from PM2.5 because it more toxic than TSP. According to your valuable comment, We added more results of NPAHs in TSP and replace Table 4 with the average values of NPAHs in two harvest seasons and average value of total samples, and also profile of NPAHs in part 3.2
Point 3:
Line 394-395. Authors mentioned that “the levels of NPAHs from RS burning in this study were in the same range with those from wood burning and slightly lower than those from coal burning “, but they did not indicate the references of those studies.
[Answer 3]:Thank you for your comment. We compared our results with the results from Yang et al., 2010. We have described that data of 1-NP and 6-NC from coal burning and wood burning were obtained from the previous report [6], weadded more information in lines 441-442 and 527-528, lines number following the track changes version)
Point 4:Regarding the comparison with motorcycle emissions, I do not think that concentrations comparison is the best option since sampling methods are quite different. I understand the point of the authors which want to make a comparison about the NPAHs 1NP and 6NC, but maybe would be more convenient the emission profile comparison. Additionally, the comparison should be between TSP since that particle size was used for motorcycles, but it seems that it was made with RS PM2.5 emissions.
[Answer 4]:We collectedparticulate matter (TSP)directly from motorcycle exhausts and smoke directly from RS burning (sampling devices is about 4-5 m far away from downwind edge of the burning paddy). Then we would like to compare the contribution of 1-NP and 6-NC in particles emitted from these fuel burning. The NPAHs data in RS burning that we compared in the previous version was in TSP. According to your valuable comment, we compared profile of NPAHs in TSP between two kind of burnings (Figure 7). Please kindly see the revised manuscript with red color.
